# The Mosquito Larvicidal Activity of Lignans from Branches of *Cinnamomum camphora* chvar. Borneol

**DOI:** 10.3390/molecules28093769

**Published:** 2023-04-27

**Authors:** Zhiyong Xu, Junhui Chen, Ruifeng Shang, Fan Yang, Chuanqi Xie, Yunfei Liu, Xuefang Wen, Jianping Fu, Wei Xiong, Lei Wu

**Affiliations:** 1Institute of Applied Chemistry, Jiangxi Academy of Sciences, Nanchang 330096, China; xuzhiyong@jxas.ac.cn (Z.X.);; 2Institute of Microbiology, Jiangxi Academy of Sciences, Nanchang 330096, China; allenchen0426@gmail.com; 3School of Pharmaceutical Sciences, Jiangxi University of Chinese Medicine, Nanchang 330096, China; 4College of Food Sciences and Engineering, Jiangxi Agricultural University, Nanchang 330096, China

**Keywords:** *Cinnamomum camphora* chvar. Borneol, lignans, mosquito control, structure–activity relationship

## Abstract

The chemical investigation of branches of *Cinnamomum camphora* chvar. Borneol guided by mosquito larvicidal activity led to the isolation of fourteen known lignans (**1**–**14**). Their structures were elucidated unambiguously based on comprehensive spectroscopic analysis and comparison with the literature data. This is the first report of these compounds being isolated from branches of *Cinnamomum camphora* chvar. Borneol. Compounds **3**–**5** and **8**–**14** were isolated from this plant for the first time. All compounds isolated were subjected to anti-inflammatory, mosquito larvicidal activity and cytotoxic activity evaluation. Compounds (**1**–**14**) showed significant mosquito larvicidal activity against *Culex pipiens quinquefasciatus* with lethal mortality in 50% (LC_50_), with values ranging from 0.009 to 0.24 μg/mL. Among them, furofuran lignans(**1**–**8**) exhibited potent mosquito larvicidal activity against *Cx. p. quinquefasciatus*, with LC_50_ values of 0.009–0.021 μg/mL. From the perspective of a structure–activity relationship, compounds with a dioxolane group showed high mosquito larvicidal activity and have potential to be developed into a mosquitocide.

## 1. Introduction

Mosquitoes transmit various diseases such as malaria, which caused 409,000 deaths in 2019 [1]. *Culex pipiens quinquefasciatus* is widely distributed south of Yangtze river in China and is known as the Japanese encephalitis virus (JEV) vector in China [2]. Because of the lack of vaccines, vector control has been considered as an effective approach to reducing mosquito-borne cases [3]. However, the extensive use of limited available chemicals has caused increasing resistance. For example, *Cx. p. quinquefasciatus* has become more or less resistant to permethrin, deltamethrin, temephos, chlorpyrifos, malathion and dieldrin in La Réunion Island [4].

*Cinnamomum camphora* chvar. Bornel is a subtropical evergreen broad-leaved tree belonging to the *Cinnamomum camphora* of Lauraceae. This species is considered to be a special chemical type of camphor tree. The volatile oil extracted from its fresh branches and leaves is rich in D-borneol (natural borneol), which is the best plant choice for obtaining natural borneol at present. However, few studies have been conducted on the chemical components other than its non-volatile oil. Previous studies found that the crude CH_2_Cl_2_ fraction obtained from the EtOH extract of branches of Cinnamomum *camphora* chvar. Borneol had excellent mosquito larvicidal activity against *Culex pipiens quinquefasciatus*. In our further searches for mosquito larvicidal active metabolites from the crude CH_2_Cl_2_ extraction, 14 lignans were afforded. We report herein the isolation, structure elucidation and biological activity of them.

## 2. Results

### 2.1. Structure Elucidation of the Isolated Compound

The crude n-hexane, CH_2_Cl_2_, EtOAc, n-BuOH and aqueous phase fractions were obtained from the EtOH extract of branches of *Cinnamomum camphora* chvar. Borneol by extraction. Among the five extraction stages, the CH_2_Cl_2_ fraction displayed the most prominent mosquito larvicidal activity against *Cx. p. quinquefasciatus* with lethal mortality in 50% (LC_50_) values of 0.032 μg/mL. To further explore the mosquito larvicidal chemical components, the CH_2_Cl_2_ extract was subjected to silica gel chromatography, Sephadex LH-20, reversed phase column chromatography and Preparative HPLC. This led to the isolation of compounds **1**–**14** (Figure 1), including Medioresinol (**1**) [5], Syringaresinol (**2**) [6], Pinoresinol (**3**) [7], Kobusin (**4**) [8], piperitol (**5**) [9], sesamin (**6**) [10], 9(*R*)-hydroxy-d-sesamin (**7**) [11], aptosimon (**8**) [12], acuminatolide (**9**) [13], (2*R*, 3*R*)-2,3-di-(3, 4-dimethoxybenzyl)-butyrolactone (**10**) [14], (−)-Dihydro-3′,4′-dimethoxy-3′,4′-demethylenedloxycubebin (**11**) [15], balanophonin (**12**) [16], buddlenol D (**13**) [17], (7*R*, 7′*R*, 7″*S*,7‴*S*, 8*S*, 8′*S*, 8″*S*, 8‴*S*)-4″,4‴-dihydroxy-3, 3′, 3″, 3‴, 5, 5′-hexamethoxy-7, 9′; 7′, 9-diepoxy-4, 8″; 4’, 8‴-bisoxy-8 and 8′-dineolignan-7″, 7‴, 9″, 9‴-tetraol (**14**) [18]. 

Compounds **3**–**5** and **8**–**14** were first isolated from the titled plant and structures of all compounds were identified based on NMR spectroscopic methods, mass spectrometry, as well as by comparison with the literature data (See the Appendix A). Compounds **1**–**8** were furan lignans with the same basic mother nucleus and the relative configuration subtypes of compounds **2**–**6** were decided as trans-(H-7, 8, 8′, 9) by the shift difference between two protons at position 7′ and the shift difference between two protons at position 9 (Δδ_H-7′_ and Δδ_H-9_ =0.3~0.4) [19]. More specifically, the structural differences of compounds **1**–**3** were reflected in the substitution of hydroxyl and methoxy groups on the benzene ring. Compounds **4**–**5** and **6**–**8** differed structurally from compounds **1**–**3** in that they contained one or two methylenedioxy groups. Different similar compounds with different substituents may have led to different biological activities.

### 2.2. Mosquito Larvicidal Activity of Lignans(**1**–**14**)

All compounds isolated were subjected to mosquito larvicidal activity evaluation. As shown in Table 1, compounds (**1**–**14**) showed significant mosquito larvicidal activity against *Cx. p. quinquefasciatus*, with LC_50_ values ranging from 0.009 to 0.24 μg/mL. No mortality was observed in the DMSO-treated group but permethrin (positive control) showed high toxicity with an LC_50_ value of 0.007 μg/mL. Among them, furofuran lignans (**1**–**8**) exhibited potent mosquito larvicidal activity against *Cx. p. quinquefasciatus*, with LC_50_ values of 0.009–0.021 μg/mL. In particular, kobusin (**4**), piperitol (**5**), sesamin (**6**) and aptosimon (**8**) exhibited comparable mosquito larvicidal activity against *Cx. p. quinquefasciatus* to that of the positive control, with LC_50_ values of 0.01, 0.009, 0.01 and 0.011, respectively. Other furan lignans, including Medioresinol (**1**), Syringaresinol (**2**), Pinoresinol (**3**), 9(*R*)-hydroxy-d-sesamin (**7**), acuminatolide (**9**), buddlenol D (**13**) and (7*R*, 7′*R*, 7″*S*, 7‴*S*, 8*S*, 8′*S*, 8″*S*, 8‴*S*)-4″,4‴-dihydroxy-3, 3’, 3″, 3‴, 5, 5′-hexamethoxy-7, 9′; 7′, 9-diepoxy-4, 8″; 4’,8‴-bisoxy-8, 8’-dineolignan-7″, 7‴, 9″, 9‴-tetraol (**14**) exhibited slightly weaker anti-mosquito activity compared to the above compounds, with LC_50_ values of 0.02, 0.021, 0.021, 0.016, 0.047, 0.039, 0.039, respectively. The mosquito larvicidal activity against *Cx. p.* quinquefasciatus of (2*R*, 3*R*)-2,3-di-(3, 4-dimethoxybenzyl)-butyrolactone (**10**), (−)-Dihydro-3′,4′-dimethoxy-3′,4′-demethylenedl-oxycubebin (**11**) and balanophonin (**12**) showed a steep decrease compared to furan lignans, with LC_50_ values of 0.106, 0.240, 0.185, respectively.

### 2.3. Anti-Inflammatory Activity, Cytotoxic Activity and Evaluation of Lignans(**1**–**14**)

All compounds isolated were subjected to anti-inflammatory and cytotoxic activity evaluation. Unfortunately, as shown in Figure 2 and Figure 3, all compounds did not show significant NO inhibitory activity on lipopolysaccharide-induced RAW 264 cells (compared with the LPS induction group, the inhibition rate was <50% at 10 μM) or cytotoxicity against the human tumor cell HepG2 (with an inhibition rate of <50% at 50 μM).

## 3. Discussion

Compounds **1**–**9**, **13** and **14** are furofuran lignans that feature with a bicyclic oxygen skeleton; these mainly showed antioxidant, insecticidal, inhibitory activity against AChE and NO production in LPS-treated BV-2 microglial cells [20]. Among them, Sesamin (**7**) has the effect of lowering cholesterol in clinical applications [20,21]. Two dibenzylbutane lignans (**10**–**11**) were also isolated from *Virola Venosa*, but their biological activity has been poorly reported [15]. Benzodihydrofuran neolignan balanophonin (**12**) was first isolated from the plant *Balanophora japonica* Makino and has obvious PGI2 induction activity [22].

Compounds **1**–**14** showed broad mosquito larvicidal activity against *Cx. p. quinquefasciatus* with LC_50_ values ranging from 0.009 to 0.24 μg/mL. Similarly, lignans have proven potential in mosquito control. Leptostachyol acetate was found to be lethal to *Culex pipiens pallens*, *Aedes aegypti* and *Ocheratatos togoi* [23]; haedoxan A exhibited high activity against *Aedes aegypti* larvae [24]; Phrymarolin-I, haedoxane A and haedoxane E were toxic to *Cx. p. pallens* [25]; and (+)-xanthoxylol-γ,γ-dimethylallylether (XDA) showed ability against *Culex pipiens pallens* and *Aedes aegypti* [26].

In a sense, as the main characteristic component of the branches of *Cinamomum camphora* chvar. Borneol, lignans may play a certain role in ecology such as protecting themselves from mosquitoes and pests. Among them, furofuran lignans(**1**–**8**) exhibited potent mosquito larvicidal activity against *Cx. p. quinquefasciatus*, with LC_50_ values of 0.009–0.021 μg/mL; these values are far stronger than compounds dibenzylbutane lignans (**10**–**11**) and benzodihydrofuran neolignan balanophonin (**12**), thus indicating the presence of a dioxolane group in compounds enhancing mosquito larvicidal activity. From the perspective of a structure–activity relationship, the mosquito larvicidal activity against *Cx. p. quinquefasciatus* in comparison to compounds **1**–**3**, **4** and **6**, **6**–**8** shows that there is no effect on the methoxy substitutions at the 3 and 3′ sites of the benzene ring. There is also no effect on whether the 3′ and 5′ methoxy groups form a ring, but the hydroxyl substitution and the formation of double bonds at position 9 have a significant effect. It is necessary to conduct further research on the resistance of lignans to mosquitoes and insects, especially furan lignan analogues with a methylenedioxy group such as structural optimization. This will determine whether it is related to configuration or whether it is related to bioecology.

## 4. Materials and Methods

### 4.1. General Experimental Procedures

NMR spectra were recorded on a Bruker AV-400 (Bruker Corporation, Switzerland) instrument with TMS as an internal standard. Optical rotation was recorded at 25 °C using a WYA-2S digital Abbe polarimeter (Shanghai Physico-optical Instrument Factory). ESI-MS spectra were recorded on a VG Auto Spec-3000 mass spectrometer (VG, Manchester, UK). High-resolution ESI-MS were recorded on an Agilent 6210 mass spectrometer employing peak matching. Preparative HPLC was performed on a Waters Prep 150 equipped with a Waters 2489 UV/visible detector and XBridge BEH C18 Column (130 Å, 5 μm, 4.6 mm × 250 mm). SephadexLH-20 (GE Healthcare Bio-Sciences AB, Uppsala, Sweden), SiliaSphere C18 (SiliCycle, Quebec, QC, Canada) and Silica gel (200–300 mesh, Qingdao Marine Chemical Factory, Qingdao, China) were used for column chromatography (CC). Thin-layer chromatography (TLC) was performed on precoated silica gel GF254 plates (Qingdao Marine Chemical Factory, Qingdao, China).

### 4.2. Plant Material and Mosquitoes

The branches of *Cinnamomum camphora* chvar. Borneol were collected from Ji An (114.62° E, 27.38° N), Jiang Xi Province, People’s Republic of China in November 2020, and authenticated by Professor Xionghui Li (Jiangxi Academy of Sciences). The voucher specimen (No. PML202102) was deposited in the herbarium of Jiangxi Academy of Sciences.

*Cx. p. quinquefasciatus* were reared in an insect room, maintained at 26 ± 1 °C, 65 ± 10% relative humidity (RH) in a 12 h: 12 h (light: dark cycle) with an artificial diet of yeast (50): lactose albumin (50).

### 4.3. Extraction and Isolation

The air-dried branches of *Cinnamomum camphora* chvar. Borneol (5 kg) were ground into powder and extracted with EtOH/H_2_O (95:5, *v*:*v*, 3 × 10 L) at room temperature. The filtrates were evaporated under reduced pressure to yield a crude gum (252 g), which was dissolved in warm water (50 °C) and extracted with petroleum ether (3 × 4 L), dichloromethane (3 × 4 L), ethyl acetate (3 × 4 L), n-BuOH (3 × 4 L) successively to obtain a CH_2_Cl_2_ extract (50 g). The CH_2_Cl_2_ extract (50 g) was subjected to vacuum liquid chromatography (VLC) on silica gel using a step gradient of CH_2_Cl_2_/MeOH (100:0, 100:1, 100:2, 100:4, 10:1, *v*:*v*) to afford 5 fractions (fraction 1- fraction 5) and compound **7** (80.5 mg, colorless crystals). Fraction 1 was eluted with a step gradient of petroleum ether/ethyl acetate (10:1, 5:1, 3:1, 1:1) to give 4 subfractions (fraction 1.1- fraction 1.4). Compound **6** (10.8 mg, t_R_ 11.4 min) and compound **8** (4.0 mg, t_R_ 45.6 min) were obtained from fraction 1.1 using a preparative RP-C18 HPLC (CH_3_CN–H_2_O, 38:62). Fraction 1.2 was further purified by Sephadex LH-20 CC (CH_2_Cl_2_/MeOH, 1:1) and finally by a preparative RP-C18 HPLC (CH_3_CN–H_2_O, 37:63) to yield compound **2** (7.6 mg, t_R_ 10.4 min) and compound **9** (6.1 mg, t_R_ 16.2 min). Fraction 1.3 was purified by Sephadex LH-20 CC (MeOH) and finally by a C18 ODS column eluted with 34% MeOH-H_2_O to obtain compound **1** (4.8 mg) and compound **3** (6.7 mg). Fraction 2 was eluted with a step gradient of petroleum ether/ethyl acetate (5:1, 3:1, 1:1) to give 3 subfractions (fraction 2.1- fraction 2.3). Compound **4** (99.4 mg) and compound **10** (644 mg) were isolated from fraction 2.1 and 2.3 by recrystallization, respectively. Compound **5** (28.8 mg) was acquired from fraction 3 by a silica gel CC equivalently eluted with petroleum ether/ethyl acetate (5:1). Fraction 4 was further purified by a C18 ODS column eluted with a step gradient of 30%–100% MeOH-H_2_O, giving 4 subfractions (fraction 4.1- fraction 4.4). Compounds **12** (18.6 mg, t_R_ 55 min), **13** (5.1 mg, t_R_ 34 min), **14** (2.3 mg, t_R_ 53 min) and **11** (2.2 mg, t_R_ 45 min) were isolated from fraction 4.3 by a preparative RP-C18 HPLC (CH_3_CN–H_2_O, 25:75).

*Medioresinol* (**1**): white powder; ESl-MS *m*/*z*, 388.0[M]^+^(C_21_H_24_O_7_); ^1^H NMR(400 MHz, CDCl_3_) *δ*_H_ 6.90 (1H, d, *J* = 2.0 Hz, H-2′), 6.88 (1H, d, *J* = 2.0 Hz, H-6′), 6.82 (1H, dd, *J* = 8.1, 1.8 Hz, H-5′), 6.59 (2H, s, H-2, 6), 5.65 (1H, s, C-4 or 4′ OH), 5.54 (1H, s, C-4 or 4′ OH), 4.74 (2H, dd, *J* = 11.4, 4.4 Hz, H-7, 9′), 4.36–4.17 (2H, m, H-7′a, 9a), 3.90 (9H, s, C-3, 3’, 5-OCH_3_), 3.84–3.77 (2H, m, H-7′b, 9b), 3.16–2.99 (2H, m, H-8, 8′); ^13^C-NMR (100 MHz, CDCl_3_) *δ*_C_ 147.3 (C-3, 5), 146.9 (C-3′), 145.4 (C-4′), 134.4 (C-4), 133.0 (C-1′), 132.3 (C-1), 119.1 (C-6′), 114.4 (C-5′), 108.8 (C-2′), 102.9 (C-2, 6), 86.3 (C-7), 86.0 (C-9′), 72.0 (C-9), 71.8 (C-7′), 56.5(3, 5-OCH_3_), 56.1 (3′-OCH_3_), 54.5 (C-8), 54.2 (C-8′).

*Syringaresinol* (**2**): white powder; ESl-MS *m*/*z*, 419.1[M + H]^+^(C_22_H_27_O_8_); ^1^H NMR(400 MHz, CDCl_3_) *δ*_H_ 6.57 (4H, s, H-2, 2′, 6, 6′), 5.78 (2H, s, OH), 4.72 (2H, d, *J* = 4.2 Hz, H-7, 9′), 4.27 (2H, dd, *J* = 9.0, 6.7 Hz, H-7′a, 9a), 3.87 (2H, dd, *J* = 9.0, 3.4 Hz, H-7′b, 9b), 3.84 (12H, s, 3, 3′, 5, 5′-OCH_3_), 3.09 (2H, s, H-8, 8′); ^13^C-NMR (100 MHz, CDCl_3_) *δ*_C_ 147.1 (C-3, 3′, 5, 5′), 134.3 (C-4, 4′), 132.0 (C-1, 1′), 102.7 (C-2, 2′, 6, 6′), 86.0 (C-7, 9′), 71.7 (C-9, 7′), 56.3(3, 3′, 5, 5′-OCH_3_), 54.2 (C-8, 8′).

*Pinoresinol* (**3**): Oil; HRESl-MS *m*/*z* 359.1489[M + H]^+^(Calcd for C_20_H_23_O_6_, 359.1491); ^1^H NMR(400 MHz, CDCl_3_) *δ*_H_ 6.81–6.90 (6H, m, H-2, 2′, 5, 5′, 6, 6′), 5.66 (2H, s, OH), 4.74 (2H, d, *J* = 3.7 Hz, H-7, 9′), 4.25 (2H, dd, *J* = 8.8, 4.4 Hz, H-7′a, 9a), 3.88 (6H, s, 3, 3′-OCH_3_), 3.87 (2H, dd, *J* = 8.8, 4.4 Hz, H-7′b, 9b), 3.10 (2H, m, H-8, 8′); ^13^C-NMR (100 MHz, CDCl_3_) *δ*_C_ 146.9 (C-3, 3′), 145.4 (C-4, 4′), 133.1 (C-1, 1′), 119.1 (C-6, 6′), 114.4 (C-5, 5′), 108.8 (C-2, 2′), 86.0 (C-7, 9′), 71.8 (C-7′, 9), 56.1 (3, 3′-OCH_3_), 54.3 (C-8, 8′).

*Kobusin* (**4**): white solid; ESl-MS *m*/*z*, 371.1[M + H]^+^(C_21_H_23_O_6_); ^1^H NMR(400 MHz, DMSO-d_6_) *δ*_H_ 7.06–6.76 (6H, m, H-2, 5, 6, 2′, 5′, 6′), 5.99 (2H, s, H-10), 4.66 (2H, d, *J* = 4.8 Hz, H-7, 9′), 4.16–4.12 (2H, m, H-9a, 7′a), 3.78 (2H, d, *J* = 4.0 Hz, H-9b, 7′b), 3.76 (3H, s, 4′-OCH_3_), 3.74 (3H, s, 3′-OCH_3_), 3.10–2.94 (2H, m, H-8, 8′); ^13^C-NMR (100 MHz, DMSO-d_6_) *δ*_C_ 148.8 (C-3′), 148.2 (C-4′), 147.4 (C-3), 146.4 (C-4), 135.5 (C-1), 133.9 (C-1′), 119.3 (C-6), 118.2 (C-6′), 111.7 (C-5), 110.0 (C-5′), 107.9 (C-2), 106.5 (C-2′), 100.8 (OCH2O), 85.0 (C-7′), 84.9 (C-7), 71.0 (C-9), 70.9 (C-9′), 55.5 (3′-OCH_3_), 55.5 (4′-OCH_3_), 53.8 (C-8), 53.6 (C-8′).

*Piperitol* (**5**): white needle crystal; ESl-MS *m*/*z*, 379.2[M + Na]^+^(C_20_H_20_O_6_Na); ^1^H NMR(400 MHz, DMSO-d_6_) *δ*_H_ 6.95–6.80 (4H, m, H-2, 2′, 5, 6′), 6.79–6.67 (2H, m, H-5′, 6), 5.99 (2H, s, OCH_2_O), 4.63 (2H, dd, *J* = 9.5, 4.4 Hz, H-7, 9′), 4.19–4.04 (2H, m, H-9a, 7′a), 3.77 (3H, s, 3-OCH_3_), 3.76–3.71 (2H, m, H-9b, 7′b), 3.09–2.93 (2H, m, H-8, 8′); ^13^C-NMR (100 MHz, DMSO-d_6_) *δ*_C_ 147.5 (C-3), 147.3 (C-3′), 146.4 (C-4′), 145.9 (C-4), 135.5 (C-1′), 132.2 (C-1), 119.3 (C-6′), 118.6 (C-6), 115.1 (C-5), 110.5 (C-2), 107.9 (C-5′), 106.5 (C-2′), 100.8 (OCH_2_O), 85.1 (C-7), 84.9 (C-9′), 71.0 (C-9), 70.8 (C-7′), 55.6 (3-OCH_3_), 53.8 (C-8′), 53.5 (C-8).

*Sesamin* (**6**): white needle crystal; HRESl-MS *m*/*z* 355.1163[M + H]^+^(Calcd for C_20_H_19_O_6_, 355.1165); ^1^H NMR(400 MHz, DMSO-d_6_) *δ*_H_ 6.90 (2H, s, H-2, 2′), 6.89–6.79 (4H, m, H-5, 5′, 6, 6′), 5.99 (4H, s, H-10, 10′), 4.64 (2H, d, *J* = 3.6 Hz, H-7, 9′), 4.11 (2H, dd, *J* = 8.6, 6.7 Hz, H-9a, 7′a), 3.76 (2H, dd, J = 9.0, 2.9 Hz, H-9b, 7′b), 3.06–2.91 (2H, m, H-8, 8′); ^13^C-NMR (100 MHz, DMSO-d_6_) *δ*_C_ 147.4 (C-4, 4′), 146.4 (C-3, 3′), 135.4 (C-1, 1′), 119.3 (C-6, 6′), 107.9 (C-2, 2′), 106.5 (C-5, 5′), 100.8 (C-10, 10′), 84.8 (C-7, 9′), 70.9 (C-9, 7′), 53.7 (C-8, 8′).

*9(R)-Hydroxy-d-sesamin* (**7**): white needle crystal; ESl-MS *m*/*z*, 371.2[M + H]^+^(C_21_H_23_O_6_); ^1^H NMR(400 MHz, DMSO-d_6_) *δ*_H_ 7.12 (1H, s, H-2), 6.94–6.82 (5H, m, H-2′, 5, 5′, 6, 6′), 6.66 (1H, d, *J* = 4.8 Hz, 9-OH), 6.00 (4H, s, 2×OCH_2_O), 5.43 (1H, d, *J* = 4.6 Hz, H-9), 4.78 (2H, dd, *J* = 27.9, 6.6 Hz, H-7, 9′), 4.12 (1H, dd, *J* = 8.2, 6.4 Hz, H-7′a), 3.94 (1H, d, *J* = 8.3 Hz, H-7′b), 3.30–2.97(1H, m, H-8′), 2.69 (1H, m, H-8); ^13^C-NMR (100 MHz, DMSO-d_6_) *δ*_C_ 147.5 (C-3), 147.3 (C-3′), 146.5 (C-4), 146.3 (C-4′), 137.3 (C-1), 136.2 (C-1′), 119.4 (C-6), 119.1 (C-6′), 108.0 (C-2), 107.7 (C-2′), 106.8 (C-5), 106.2 (C-5′), 100.9 (C-9), 100.8 (OCH_2_O), 86.0 (C-7′), 82.6 (C-7), 71.4 (C-9′), 62.1 (C-8), 53.4 (C-8′).

*Aptosimon* (**8**): colorless oil; HRESl-MS *m*/*z* 369.0961[M + H]^+^(Calcd for C_20_H_17_O_7_, 369.0958); ^1^H NMR(400 MHz, DMSO-d_6_) *δ*_H_ 7.06 (1H, s, H-2), 6.96–6.82 (5H, m, H-2′, 5, 5′, 6, 6′), 6.03 (4H, d, *J* = 8.0 Hz, H-10, 10′), 5.44 (1H, d, *J* = 3.6 Hz, H-7), 5.14 (1H, d, *J* = 3.8 Hz, H-9′), 4.17 (1H, dd, *J* = 9.2, 7.4 Hz, H-7′a), 3.96 (1H, d, *J* = 9.4, 4.5 Hz, H-7′b), 3.78 (1H, dd, *J* = 9.3, 3.8 Hz, H-8), 3.31–3.24(1H, m, H-8′); ^13^C-NMR (100 MHz, DMSO-d_6_) *δ*_C_ 176.8 (C-9), 147.7 (C-3), 147.5 (C-3′), 147.4 (C-4), 146.8 (C-4′), 134.2 (C-1), 133.5 (C-1′), 120.1 (C-6), 119.5 (C-6′), 108.1 (C-2), 108.0 (C-2′), 106.6 (C-5), 106.4 (C-5′), 101.2 (C-10), 101.0 (C-10′), 84.3 (C-7), 82.7 (C-9′), 72.2 (C-7′), 52.3 (C-8), 48.6 (C-8′).

*Acuminatolide* (**9**): white needle crystal; HRESl-MS *m*/*z* 249.0757[M + H]^+^(Calcd for C_13_H_13_O_5_, 249.0753); ^1^H NMR(400 MHz, DMSO-d_6_) *δ*_H_ 6.96 (1H, s, H-2), 6.87 (2H, s, H-5, 6), 6.00 (2H, s, H-10), 4.69 (1H, d, *J* = 6.3 Hz, H-7), 4.48 (1H, dd, *J* = 9.4, 6.9 Hz, H-9′a), 4.34 (1H, dd, *J* = 9.5, 1.8 Hz, H-7′a), 4.18 (1H, t, *J* = 8.6 Hz, H-9′b), 3.95 (1H, dd, *J* = 9.0, 3.2 Hz, H-7′b), 3.57 (1H, td, *J* = 8.7, 3.2 Hz, H-8′), 3.09 (1H, dtd, *J* = 8.6, 6.8, 1.8 Hz, H-8); ^13^C-NMR (100 MHz, DMSO-d_6_) *δ*_C_ 178.5 (C-9), 147.5 (C-3), 146.8 (C-4), 134.1 (C-1), 119.5 (C-6), 108.0 (C-5), 106.5 (C-2), 101.0 (C-10), 85.3 (C-7), 70.1 (C-9′), 69.5 (C-7′), 47.6 (C-8′), 45.8 (C-8).

*(2R, 3R)-2,3-Di-(3, 4-dimethoxybenzyl)-butyrolactone* (**10**): white needle crystal; HRESl-MS *m*/*z* 409.1621[M + Na]^+^(Calcd for C_22_H_26_O_6_Na, 409.1638); ^1^H NMR(400 MHz, DMSO-d_6_) *δ*_H_ 6.95–6.39 (6H, m, H-2′, 5′, 6′, 2″, 5″, 6″), 4.10 (1H, d, *J* = 7.2 Hz, H-4a), 3.88 (1H, d, *J* = 7.2 Hz, H-4b), 3.71 (12H, s, 3′, 4′, 3″, 4″-OCH_3_), 2.82 (2H, dt, *J* = 20.0, 11.3 Hz, H-7), 2.70 (1H, d, *J* = 5.0 Hz, H-2), 2.57 –2.36 (3H, m, H-3, 5); ^13^C-NMR (100 MHz, DMSO-d_6_) *δ*_C_ 178.3 (C-1), 148.7 (C-3′), 148.6 (C-3″), 147.5 (C-4′), 147.4 (C-4″), 131.2 (C-1′), 130.6 (C-1″), 121.2 (C-6′), 120.4 (C-6″), 113.2 (C-2′), 112.5 (C-2″), 111.9 (C-5′), 111.8 (C-5″), 70.7 (C-4), 55.4 (3′, 4′, 3″, 4″-OCH_3_), 45.6 (C-2), 40.8 (C-3), 36.9 (C-7), 33.7 (C-5).

*(−)-Dihydro-3′,4′-dimethoxy-3′,4′-demethylenedloxycubebin* (**11**): White powder; [α]^25^_D_-10.2 (c 0.36, CHCl_3_); HRESl-MS *m*/*z* 375.1802[M + H]^+^(Calcd for C_21_H_27_O_6_, 375.1805); ^1^H NMR(400 MHz, DMSO-d_6_) *δ*_H_ 6.84–6.53 (6H, m, H-2, 5, 6, 2′, 5′, 6′), 5.95 (2H, s, O_2_CH_2_), 4.55 (2H, d, *J* = 4.1 Hz, 9, 9′-OH), 3.71 (3H, s, 3′-OCH_3_), 3.68 (3H, s, 4′-OCH_3_), 3.44 –3.36 (4H, m, H-9, 9′), 2.60–2.42 (2H, m, H-7, 7′), 1.92–1.72 (2H, m, H-8, 8′); ^13^C-NMR (100 MHz, DMSO-d_6_) *δ*_C_ 148.5 (C-3′), 147.0 (C-3), 146.8 (C-4′), 145.0 (C-4), 135.3 (C-1), 133.9 (C-1′), 121.7 (C-6), 120.8 (C-6′), 112.6 (C-5′), 111.7 (C-2′), 109.2 (C-5), 107.7 (C-2), 100.5 (O_2_CH_2_), 60.2 (C-9), 60.1 (C-9′), 55.5 (3′-OCH_3_), 55.3 (4′-OCH_3_), 42.7(C-8), 42.4 (C-8′), 34.0 (C-7), 33.9 (C-7′).

*Balanophonin* (**12**): Pale yellow powder; HRESl-MS *m*/*z* 357.1332[M + H]^+^(Calcd for C_20_H_21_O_6_, 357.1324); ^1^H NMR(400 MHz, DMSO-d_6_) *δ*_H_ 9.60 (1H, d, *J* = 7.8 Hz, H-9′), 9.07 (1H, s, 4-OH), 7.65 (1H, d, *J* = 15.7 Hz, H-7′), 7.32 (2H, s, H-2′, 6′), 6.92 (1H, s, H-2), 6.80 (1H, d, *J* = 7.8 Hz, H-8′), 6.77 (1H, s, H-5), 6.75 (1H, s, H-6), 5.56 (1H, d, *J* = 6.7 Hz, H-7), 5.08 (1H, t, *J* = 5.0 Hz, 9-OH), 3.84 (3H, s, 3′-OCH_3_), 3.75 (3H, s, 3-OCH_3_), 3.73–3.62 (2H, m, H-9), 3.53 (1H, dd, *J* = 12.1, 6.0 Hz, H-8); ^13^C-NMR (100 MHz, DMSO-d_6_) *δ*_C_ 194.0 (C-9′), 153.9 (C-7′), 150.7 (C-4′), 147.6 (C-3), 146.6 (C-4), 144.1 (C-3′), 131.7(C-1), 130.1 (C-5′), 127.7 (C-1′), 126.1 (C-8′), 118.9 (C-6′), 118.7 (C-6), 115.4 (C-5), 112.6 (C-2′), 110.5 (C-2), 88.1 (C-7), 62.7(C-9), 55.8 (3′-OCH_3_), 55.7 (3-OCH_3_), 52.4 (C-8).

*Buddlenol D* (**13**): Colorless oil; HRESl-MS *m*/*z* 667.2348[M + Na]^+^(Calcd for C_33_H_40_O_13_Na, 667.2352); ^1^H NMR(400 MHz, DMSO-d_6_) *δ*_H_ 8.27 (1H, s, 4′-OH), 8.13 (1H, s, 4″-OH), 6.65 (2H, s, H-2′, 6′), 6.60 (4H, s, H-2, 6, 2″, 6″), 5.17–5.09 (1H, m, H-7″), 4.81 (1H, dd, *J* = 7.8, 4.6 Hz, H-8″), 4.64 (2H, dd, *J* = 16.4, 3.7 Hz, H-7, 9′), 4.21–4.09 (3H, m, H-9, 7′a), 4.01 (1H, dd, *J* = 10.0, 5.8 Hz, H-7′b), 3.77 (6H, s, 3′, 5′-OCH_3_), 3.75 (6H, s, 3, 5-OCH_3_), 3.73 (6H, s, 3″, 5″-OCH_3_); ^13^C-NMR (100 MHz, DMSO-d_6_) *δ*_C_ 152.6 (C-3′, 5′), 147.9 (C-3, 5), 147.4 (C-3″, 5″), 136.8 (C-1′), 134.9 (C-4), 134.9 (C-4′), 134.3 (C-4″), 132.4 (C-1), 131.4 (C-1″), 104.3 (C-2″), 103.7 (C-2, 2′, 6, 6′), 103.3 (C-6″), 86.2 (C-8″), 85.3 (C-9′), 85.1 (C-7), 72.4 (C-7″), 71.2 (C-7′), 71.1 (C-9), 59.9(C-9″), 56.0 (3′, 5′-OCH_3_), 56.0 (3, 5-OCH_3_), 55.9(3″, 5″-OCH_3_), 53.7(C-8′), 53.6 (C-8).

(7*R*, 7′*R*, 7″*S*, 7‴*S*, 8*S*, 8′*S*, 8″*S*, 8‴*S*)-4″,4‴-Dihydroxy-3, 3’, 3″, 3‴, 5, 5′-hexamethoxy-7, 9’; 7′, 9-diepoxy-4, 8″; 4’, 8‴-bisoxy-8, 8’-dineolignan-7″, 7‴, 9″, 9‴-tetraol (14): White powder; HRESl-MS *m*/*z* 833.2954[M + Na]^+^(Calcd for C_42_H_50_O_16_Na, 833.2948); ^1^H NMR(400 MHz, DMSO-d_6_) *δ*_H_ 8.78 (2H, s, 4″, 4‴-OH), 6.92 (2H, s, H-6″, 6‴), 6.76–6.67 (4H, m, H-2″, 2‴, 3″, 3‴), 6.64 (s, H-2, 6, 2′, 6′), 5.10 (2H, d, *J* = 2.5 Hz, H-7″, 7‴),4.80 (2H, s, 7″, 7‴-OH), 4.67 (2H, d, *J* = 3.4 Hz, H-7, 9′), 4.23–4.16 (2H, m, H-8″, 8‴), 4.14– 4.07 (4H, m, H-9″, 9″), 4.03 (2H, d, *J* = 3.0 Hz, H-9a, 7′a), 3.85–3.79 (2H, m, H-9b, 7′b), 3.75 (18H, d, *J* = 8.4 Hz, 3, 3′, 3″, 5, 5′, 5″-OCH_3_), 3.12–2.99 (2H, m, H-8, 8′); ^13^C-NMR (100 MHz, DMSO-d_6_) *δ*_C_ 152.6 (C-3′, 5′), 152.6 (C-3, 5), 146.9 (C-5″, 5‴), 145.3 (C-4″, 4‴), 136.8 (C-4, 4′), 134.8 (C-1, 1′), 133.3 (C-1″, 1‴), 119.4 (C-2″, 2‴), 114.6 (C-3″, 3‴), 111.0 (C-6″, 6‴), 103.3 (C-2, 6, 2′, 6′), 86.1 (C-8″, 8‴), 85.1 (C-7, 9′), 72.1 (C-7″, 7‴), 71.3 (C-7′, 9), 59.8 (C-9″, 9‴), 56.0 (3, 3′, 5, 5′-OCH_3_), 55.5(5″, 5‴-OCH_3_), 53.6 (C-8, 8′).

### 4.4. Biological Assays

Larvicidal bioassays were conducted based on the WHO requirement with slight modification [27]. Serial concentrations (10, 20, 40, 60, 80 and 100 mg/L) were tested for lignans. Thirty 4th instar larvae were tested in a 150 mL glass beaker with 100 mL of sterilized water and 5 replicates were conducted. Mortality was recorded after 24 h of treatment, and no food was provided during the treatment. Dimethyl sulfoxide (DMSO) was set as the negative control and permethrin was set as the positive control. The antitumor activity of tested compounds against HepG2 was performed by the MTT method [28]. The anti-inflammatory activity was evaluated by the inflammatory model of LPS-induced RAW264.7 macrophages [29].

### 4.5. Statistic Analysis

SPSS (version 19.0) was used to perform the statistical analyses. Standard probit analysis was conducted for the *Cx. p. quinquefasciatus* larvicidal bioassay and LC_50_ values were calculated after 24 h of exposure. Significant differences in LC_50_ values (*p* ≤ 0.05) were concluded only if there was no overlap in the confidence intervals.

## 5. Conclusions

Fourteen known lignans including eleven furofuran lignans (**1**–**9**, **13**–**14**), two dibenzylbutane lignans (**10**–**11**) and a benzodihydrofuran neolignan (**12**) were first identified from branches of *Cinnamomum camphora* chvar. Borneol. Compounds **3**–**5** and **8**–**15** were isolated from this plant for the first time. Furofuran lignans (**1**–**9**, **13**–**14**) were found to exhibit broad mosquito larvicidal activity against *Culex pipiens quinquefasciatus*, with LC_50_ values ranging from 0.009 to 0.24 μg/mL. These results suggest that it may be meaningful to conduct complementary and further studies on lignans, especially furan lignans, in mosquito repellents and plant ecology.

## Figures and Tables

**Figure 1 molecules-28-03769-f001:**
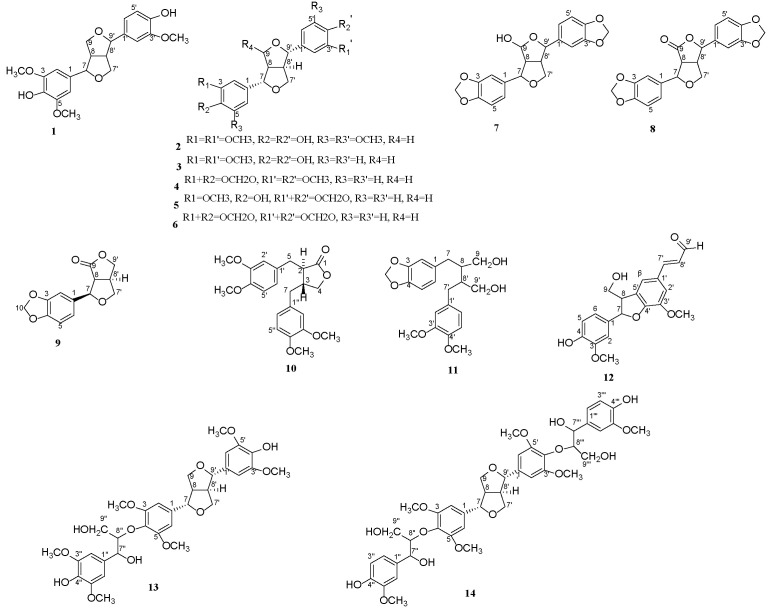
Chemical structures of compounds **1**–**14.**

**Figure 2 molecules-28-03769-f002:**
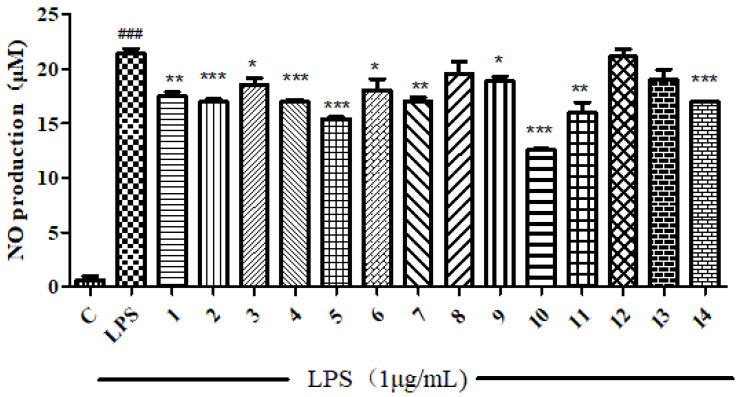
The ability to inhibit LPS-induced NO production in RAW 264 cells of compounds **1**–**14**. *** means the significant difference at *p* < 0.001, ** means *p* < 0.01, * means *p* < 0.05 compared compounds **1**–**14** with LPS. ### means *p* < 0.001 compared LPS with Control.

**Figure 3 molecules-28-03769-f003:**
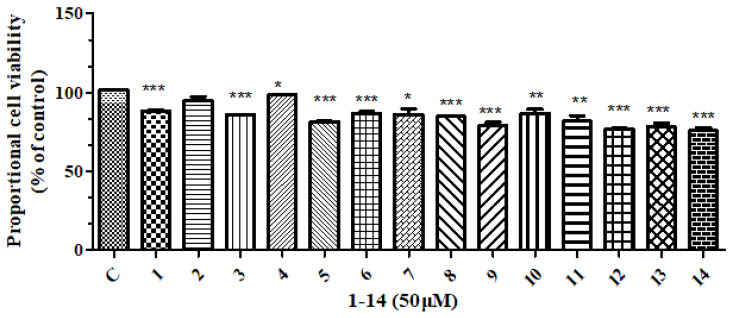
The effect of compounds **1**–14 on the HepG2 cell viability at 50 μM. *** means *p* < 0.001, ** means *p* < 0.01, * means *p* < 0.05 compared compounds **1**–**14** with Control.

**Table 1 molecules-28-03769-t001:** Effects of the CH_2_Cl_2_ fraction and compounds **1**–**14** against *Culex pipiens quinquefasciatus*. LC_50_ values (concentrations that caused mortality in 50 % of a sample population) were determined at 24 h.

Chemical	Regression Equation	*x*^2^ Value	*p*	LC_50_	95%CL
the CH_2_Cl_2_ fraction	y = 5.898 + 3.938x	75.112	0	0.032	0.027~0.037
Medioresinol (**1**)	y = 4.247 + 2.506x	13.579	0.916	0.02	0.018~0.023
Syringaresinol (**2**)	y = 4.129 + 2.455x	13.049	0.9.32	0.021	0.018~0.023
Pinoresinol (**3**)	y = 3.807 + 2.260x	26.18	0.244	0.021	0.018~0.023
Kobusin (**4**)	y = 5.057 + 2.529x	8.936	0.994	0.01	0.008~0.012
piperitol (**5**)	y = 3.824 + 1.869x	15.116	0.857	0.009	0.007~0.011
sesamin (**6**)	y = 3.951 + 1.972x	15.797	0.826	0.01	0.008~0.012
9(*R*)-hydroxy-d-sesamin (**7**)	y = 3.645 + 2.032x	43.046	0.005	0.016	0.012~0.020
aptosimon (**8**)	y = 3.468 + 1.761x	11.037	0.974	0.011	0.008~0.013
acuminatolide (**9**)	y = 2.791 + 2.108x	30.048	0.117	0.047	0.041~0.055
(2*R*, 3*R*)-2,3-di-(3, 4- dimethoxybenzyl)- butyrolactone (**10**)	y = 1.125 + 1.153x	36.884	0.024	0.106	0.079~0.161
(−)-Dihydro-3′,4′- dimethoxy-3′,4′-demethylenedloxycubebin (**11**)	y = 0.919 + 1.483x	21.332	0.500	0.240	0.183~0.355
balanophonin (**12**)	y = 1.294 + 1.767x	26.693	0.223	0.185	0.151–0.243
buddlenol D (**13**)	y = 4.520 + 3.204x	26.321	0.238	0.039	0.036~0.042
(7*R*, 7′*R*, 7″*S*, 7‴*S*, 8*S*, 8′*S*, 8″*S*, 8‴*S*)-4″,4‴-dihydroxy-3, 3′, 3″, 3‴, 5, 5′-hexamethoxy-7, 9′; 7′, 9-diepoxy-4, 8″; 4’, 8‴-bisoxy-8, 8’-dineolignan-7″, 7‴, 9″, 9‴-tetraol (**14**)	y = 4.285 + 3.050x	41.695	0.007	0.039	0.034~0.045
Permethrin (positive control)	y = 4.105 + 1.908x	10.023	0.986	0.007	0.005~0.009

## Data Availability

Data are available by request through the authors.

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
