# Peer review of "The Mosquito Larvicidal Activity of Lignans from Branches of Cinnamomum camphora chvar. Borneol"

_molecules, 2023, doi:10.3390/molecules28093769_

Round 1
Reviewer 1 Report
1. Add secondary titles to each part of the experimental results and summarize them separately.
2. Line 81-86 Move to the introduction section.
3. Line 89 Change "," to ".".
4. Line 87-89 Please add references.
5. The discussion was particularly short, far from sufficient.
6. Lack of the section Statistical Analysis.
7. Line 252 Lack of positive control.
Reviewer 2 Report
In this work, authors have obtained fourteen known lignans (1-14) from branches of Cinnamomum camphora chvar. Compounds 3-5 and 8-14 were isolated from this plant for the first time. All compounds isolated were subjected to anti-inflammatory, mosquito larvicidal activities and cytotoxic activities evaluation. Compounds (1-14) showed significant mosquito larvicidal activities against Culex pipiens quinquefasciatus with LC50 values ranging from 0.009-0.24μg/mL. Among them, furofuran lignans(1-8) exhibited potent mosquito larvicidal activities against Cx. p. quinquefasciatus, with LC50 values of 0.009-0.021μg/mL. The work is good, but some issues need to be revised before acceptance.
1. The HRMS of compounds 1-14 should be provided.
2. The data of Antitumor and anti-inflammatory activities should be provided and discussed in the text.
Reviewer 3 Report
Dear authors,
This study reported 14 known lignans from Cinnamomum camphora and their mosquito larvicidal activities. However, I have some enquiries for the authors about the structure elucidation.
i) Compounds 1-14 are a group of lignans with one or more stereocenters; however, the authors assigned the structures based on literature data. It might not be conclusive enough. Please compare all NMR data (1H and 13C) with the literature values in a table format and include these in the supplementary data.
ii) Please briefly explain how you assign the absolute or relative configuration (R or S) for these compounds (1-14) in a paragraph, for example, using NOESY correlation or CD.
iii) The authors indicated the optical rotation (+) and (-) for compounds (1), (2), (5), (8), and (11). Please include the RAW DATA of the optical rotation for these compounds.
iv) Please include the MS data for all the reported compounds.
v) What standard drug is used for mosquito larvicidal activities?
Reviewer 4 Report
The submitted article is an example of a classic phytochemical study, conducted with rigor and valuing nuclear magnetic resonance data. Literature citation was very well conducted going to the original data of the first isolation of each of the lignoids described in this work. The only weak point was the absence of the mass spectrum signal in the spectral description (methodology). Even using low-resolution Varian equipment, it would be important to report the type of ion observed. In the rest, I can only congratulate the authors for the beautiful work.
Round 2
Reviewer 1 Report
1. Part IV lacks a general title.
2. On line 317,why not 1-9?
Author Response
Question 1: Part IV lacks a general title.
Answer: Thank you! We have added a general title in Part IV. As follow:
- Materials and Methods
Question 2: On line 317,why not 1-9?
Answer: Thank you! We have have changed "1-8, 9" to "1-9" in our article.
Reviewer 2 Report
The authors have revised the manuscript well and the revised manuscript can be accepted in current form.
Author Response
Thank you very much for your recognition of the article!
Reviewer 3 Report
Thank you for the correction and clarification. The authors made the necessary changes as recommended. The NMR description and MS supporting data are sufficient enough for publication.
Author Response

(The authors gave the same response as above.)
